# Sympatholytic Mechanisms for the Beneficial Cardiovascular Effects of SGLT2 Inhibitors: A Research Hypothesis for Dapagliflozin’s Effects in the Adrenal Gland

**DOI:** 10.3390/ijms22147684

**Published:** 2021-07-19

**Authors:** Anastasios Lymperopoulos, Jordana I. Borges, Natalie Cora, Anastasiya Sizova

**Affiliations:** Laboratory for the Study of Neurohormonal Control of the Circulation, Department of Pharmaceutical Sciences, Nova Southeastern University, Fort Lauderdale, FL 33328-2018, USA; jb3837@mynsu.nova.edu (J.I.B.); nc1174@mynsu.nova.edu (N.C.); as4532@mynsu.nova.edu (A.S.)

**Keywords:** adrenal medulla, catecholamine, dapagliflozin, free fatty acid, g protein-coupled receptor kinase-2, heart failure, SGLT2 inhibitor, sympathetic neuron, sympatholysis, signal transduction

## Abstract

Heart failure (HF) remains the leading cause of morbidity and death in the western world, and new therapeutic modalities are urgently needed to improve the lifespan and quality of life of HF patients. The sodium-glucose co-transporter-2 (SGLT2) inhibitors, originally developed and mainly indicated for diabetes mellitus treatment, have been increasingly shown to ameliorate heart disease, and specifically HF, in humans, regardless of diabetes co-existence. Indeed, dapagliflozin has been reported to reduce cardiovascular mortality and hospitalizations in patients with HF and reduced ejection fraction (HFrEF). This SGLT2 inhibitor demonstrates these benefits also in non-diabetic subjects, indicating that dapagliflozin’s efficacy in HF is independent of blood glucose control. Evidence for the effectiveness of various SGLT2 inhibitors in providing cardiovascular benefits irrespective of their effects on blood glucose regulation have spurred the use of these agents in HFrEF treatment and resulted in FDA approvals for cardiovascular indications. The obvious question arising from all these studies is, of course, which molecular/pharmacological mechanisms underlie these cardiovascular benefits of the drugs in diabetics and non-diabetics alike. The fact that SGLT2 is not significantly expressed in cardiac myocytes (SGLT1 appears to be the dominant isoform) adds even greater perplexity to this answer. A variety of mechanisms have been proposed over the past few years and tested in cell and animal models and prominent among those is the potential for sympatholysis, i.e., reduction in sympathetic nervous system activity. The latter is known to be high in HF patients, contributing significantly to the morbidity and mortality of the disease. The present minireview first summarizes the current evidence in the literature supporting the notion that SGLT2 inhibitors, such as dapagliflozin and empagliflozin, exert sympatholysis, and also outlines the main putative underlying mechanisms for these sympatholytic effects. Then, we propose a novel hypothesis, centered on the adrenal medulla, for the sympatholytic effects specifically of dapagliflozin. Adrenal medulla is responsible for the production and secretion of almost the entire amount of circulating epinephrine and of a significant percentage of circulating norepinephrine in the human body. If proven true experimentally, this hypothesis, along with other emerging experimental evidence for sympatholytic effects in neurons, will shed new light on the pharmacological effects that mediate the cardiovascular benefits of SGLT2 inhibitor drugs, independently of their blood glucose-lowering effects.

## 1. Introduction

Sodium-glucose co-transporter-2 (SGLT2) inhibitors, also known as gliflozins, are drugs that inhibit this secondary active symporter isoform in the kidney to block reabsorption of glucose from the renal filtrate back into the renal tubule cell and ultimately into the blood. Thus, they lower blood glucose levels and cause glucosuria [1]. They were developed for type 2 diabetes mellitus (T2DM) treatment, but they are now well-established to also exert cardiovascular benefits in T2DM patients [2,3]. Several medications of this class have been approved or are currently under development, the most prominent of which being canagliflozin (Figure 1), the first member of this class to be FDA-approved in 2013, dapagliflozin (Figure 1), which is also used (together with insulin) for T1DM treatment, empagliflozin (Figure 1), which has the highest selectivity for SGLT2 vs. SGLT1 among the agents in this class, ertugliflozin, and sotagliflozin, which is essentially a non-specific dual SGLT1/2 inhibitor. There are additional agents approved and marketed outside the US, such as ipragliflozin, luseogliflozin, remogliflozin, and tofogliflozin.

### 1.1. Chemistry & Structure-Activity Relationships of SGLT2 Inhibitors

As far as the drug class’ chemistry and structure–activity relationships are concerned, the pharmacophore (phlorizin) is a β-D-glucoside first isolated from the bark of the apple tree. Development of these drugs was based on this natural compound. Phlorizin consists of a glucose moiety linked to a system of two phenyl rings (the “aglycone” part) connected via an ethylene bridge [4]. As glucosides are readily broken down (hydrolyzed) by gastrointestinal glucosidases, phlorizin was a poor drug candidate, and thus served only as the lead compound for the development of gliflozins. The design of these drugs mainly involved the replacement of the O-glucoside in phlorizin with a C-glucoside (Figure 1), which made them both metabolically stable (protected against hydrolysis) and more selective for SGLT2 vs. SGLT1 [5]. The structural differences among the three most common SGLT2 inhibitors (dapagliflozin, empagliflozin, and canagliflozin) are relatively subtle (Figure 1). The basic difference is that canagliflozin has a thienyl moiety bridging the two phenyl rings of the pharmacophore aglycone, whereas dapagliflozin and empagliflozin have a methylene group instead (Figure 1). The latter two also contain a chlorine substitution on their phenyl rings, whose electronegativity (i.e., “pulling” electron pairs from the aromatic π ring electron cloud) further protects against its metabolic degradation [6]. The chlorine atom also increases the inhibitory potency against SGLT2 [5,6]. In contrast, canagliflozin has a fluorine substitution on one of its phenyl rings, which is also very electronegative and increases metabolic stability [6,7]. Finally, empagliflozin, unlike canagliflozin and dapagliflozin, has one additional tetrahydrofuran ring connected via an ether bond to the distant phenyl ring of its pharmacophore (Figure 1).

### 1.2. Cardiovascular Benefits of SGLT2 Inhibitors

In addition to blood glucose lowering, SGLT2 inhibitors have been shown to reduce atherosclerosis-related events, hospitalizations for heart failure (HF), and cardiovascular as well as all-cause mortality [8,9,10,11,12,13]. In this minireview, we outline the potential molecular and physiological mechanisms underlying the cardiovascular benefits of SGLT2 inhibitors and summarize the evidence that exists in support of their sympatholytic effects, as well as potential signaling mechanisms that could mediate such effects by this class of drugs. Finally, we propose a research hypothesis of our own for a potential anti-adrenergic action of dapagliflozin in the adrenal medulla that could complement the effects of this SGLT2 inhibitor (and perhaps also of other members of the gliflozin class) in cardiac sympathetic nerves.

## 2. Potential Mechanisms Underlying the Cardiovascular Benefits of SGLT2 Inhibitors

In the DAPA-HF study, dapagliflozin reduced cardiovascular events and mortality in both diabetic and non-diabetic patients [14]. This suggested a blood glucose regulation-independent mechanism for the heart benefits of dapagliflozin and potentially of other SGLT2 inhibitors. Two major factors that play roles in the cardio-protection afforded by SGLT2 inhibitors appear to be hypotension (reduced blood pressure) and diuresis. Indeed, SGLT2 blockade in the renal proximal tubules leads to lowered glucose and Na+ reabsorption, which, in turn, results in natriuresis and osmotic glucoso-uresis with resultant glucosuria and hypovolemia (reduced blood volume) [15]. Interestingly, although the natriuretic effects of SGLT2 inhibition are transient, the reduction in both systolic and diastolic blood pressures is prolonged [16]. Elevated blood pressure is a major causative risk for HFrEF onset [17]. Therefore, it is very likely that the anti-hypertensive and diuretic effects of SGLT2 inhibitors contribute to their cardio-protection against HF development.

### Ketone Bodies and SGLT2 Inhibitor-Mediated Sympatholysis

The investigational SGLT2 inhibitor ipragliflozin was shown to prevent cardiac hypertrophy and fibrosis in a hypertensive and obese rat model without diabetes [18]. The drug reduced cardiac adverse remodeling and hypertension without affecting blood glucose, insulin levels, body weight, or heart rate to a significant extent [18]. Several studies have documented ketone body level upregulation induced by SGLT2 inhibitor treatment in T2DM patients [19,20,21,22,23,24]. Ketone bodies are also elevated specifically inside the myocardium of HF patients [25]. Therefore, it appears that SGLT2 inhibitors shift cardiac energy metabolism and efficiency by switching nutrient utilization from lipids/glucose to ketone bodies (hydroxy-fatty acids). The healthy myocardium mainly uses fatty acids (~70%) and, to a lesser extent, glucose (~30%) as energy sources [26]. Ketone bodies like β-hydroxybutyrate (3-hydroxybutyrate) and acetoacetate can also be used for energy [26]. Ketone bodies are synthesized from fatty acids in the liver and are normally circulating at low concentrations in the blood [26]. However, under conditions of very low nutrient supply (e.g., during prolonged fasting or starvation), the liver converts free fatty acids (FFAs) to these ketone bodies, so circulating acetoacetate and β-hydroxybutyrate levels are heightened and can become a major source for energy production by the myocardium [26]. Higher amounts of adenosine triphosphate (ATP), i.e., energy, are produced per molecule of oxygen consumed when a ketone body is catabolized for energy production instead of glucose or a FFA [27]. This means that cardiac energy efficiency is improved [27]. In the failing myocardium, carbohydrate uptake is suppressed in favor of elevated ketone bodies’ (mainly β-hydroxybutyrate) uptake [28]. Plasma BNP levels and cardiac ketone body uptake correlate significantly, indicating increased utilization of ketone bodies in HF [28]. Therefore, increases in circulating ketone body levels conferred by SGLT2 inhibitor treatment may be cardioprotective in diabetic HF subjects. Whether this applies to non-diabetic HF patients remains to be elucidated in future studies. Of note, and as explained in detail in the following segment, ketone bodies are by no means biologically inert molecules useful only as energy fuel sources. They also play major roles as signaling molecules inside cells. For instance, β-hydroxybutyrate can act as a histone deacetylase (HDAC) inhibitor, affecting gene expression and, in turn, cardiac adverse remodeling [29]. Given that HDAC inhibition improves function and metabolism, while suppressing hypertrophy and inflammation, of the failing heart, it is quite plausible that inhibition of cardiac HDACs by β-hydroxybutyrate ameliorates HF progression [30]. This is in addition to this ketone body’s role as antagonist of the endogenous G protein-coupled receptor (GPCR) 41 (GPR41), which is a FFA receptor (specifically FFA receptor type 3, FFAR3). As outlined in more detail below, this has major implications for sympathetic never activity regulation.

## 3. Current Evidence for SGLT2 Inhibitor-Induced Sympatholysis

### 3.1. Evidence from Clinical Trials and Animal Models

Empagliflozin treatment leads to blood pressure and plasma volume decreases, but also, somewhat unexpectedly, to heart rate reduction in patients treated with this SGLT2 inhibitor [16]. Consistent with this finding from the EMPA-REG OUTCOME clinical trial, another SGLT2 inhibitor, luseogliflozin, also causes bradycardia in patients with relatively elevated heart rate [31]. This heart rate-reducing effect of SGLT2 inhibitors is suggestive of sympathetic (adrenergic) nervous system activity suppression, i.e., sympatholysis. Sympathetic nervous system hyperactivity is intricately linked with initiation, progression, and deterioration (poor prognosis) of chronic human HFrEF [32]. For this reason, β-blockers are nowadays part of the cornerstone pharmacotherapy regimen for HF patients, an idea hardly popular when first proposed by Bristow and colleagues back in the late 1980s–early 1990s [33]. Indeed, the pro-arrhythmic and oxygen demand-increasing effects of elevated circulating catecholamine hormones (norepinephrine and epinephrine) are maladaptive and toxic for the failing myocardium that no longer operates on the Frank–Starling curve of cardiac function and is thus unable to respond properly to adrenergic or other inotropic stimuli [32]. β-blockers protect the failing myocardium against adrenergic overstimulation by a) occupying the cardiac β-adrenergic receptors (ARs), thereby preventing endogenous catecholamines from activating them, and b) by directly reducing sympathetic nerve activity, neuronal firing, and, ultimately, catecholamine release, i.e., they effect sympatholysis [32,33]. If SGLT2 inhibitors, like the β-blockers, suppress sympathetic nerve activity, this can have profound impact on their beneficial effects in the failing myocardium. In that context, the effects of the gliflozins on ketone body metabolism mentioned above might play an important role in their sympatholytic and broader cardioprotective actions.

### 3.2. Role of FFAR3

As mentioned in the preceding section, FFAs are both essential nutrients for the cell’s metabolism/energy and important signaling molecules regulating various cellular processes. Five different types of orphan GPCRs, GPR40, GPR41, GPR43, GPR84, and GPR120, have been identified as being capable of binding to, and being activated by FFAs [34,35,36,37]. GPR40 is also known as FFA receptor type 1 (FFAR1), GPR43 as FFAR2, GPR41 as FFAR3, and GPR120 is also known as FFAR4 [34,35,36,37]. Long-chain fatty acids (LCFAs) serve as specific agonists for GPR40 (FFAR1) and GPR120 (FFAR4) [34,35], whereas GPR84 is activated by medium chain length FFAs [36]. On the other hand, short-chain fatty acids (SCFAs) activate FFAR3 (GPR41) and FFAR2 (GPR43) [37,38]. Stimulation of FFAR3 results in inhibition of the synthesis of the crucial second messenger cyclic 3’,5’-adenosine monophosphate (cAMP), as well as activation of the mitogen-activated protein kinases (MAPKs) ERKs (extracellular signal-regulated kinases) [39,40]. This is as FFAR3 couples mainly to the inhibitory/other Gi/o family of heterotrimeric G proteins that normally inhibit adenylyl cyclase (AC) [39,40]. AC is the membrane-bound enzyme that synthesizes cAMP [41]. FFAR3 is expressed in adipose tissue where it induces leptin release [42,43]. It is also highly expressed in sympathetic ganglia in both humans and mice [44]. A very interesting study conducted in transgenic mice a decade ago reported that the SCFA propionic acid activates FFAR3 in sympathetic ganglia augmenting sympathetic nervous system outflow [44]. In contrast, the ketone body β-hydroxybutyrate, produced, as mentioned above, during starvation or upon SGLT2 inhibitor treatment of diabetic patients, binds FFAR3 and blocks it, suppressing sympathetic nervous system activity (Figure 2) [44]. The authors went on to dissect the specific signaling pathway underlying these effects of FFAR3 in sympathetic neurons and discovered that FFAR3 elicited activation of sympathetic neuronal firing by inducing a non-canonical signaling cascade involving the free Gβγ subunits of activated Gi/o proteins, to which this receptor couples [44]. Free Gβγ subunits stimulate, in turn, phospholipase C (PLC)-β, which induces MAPK (ERK) signaling [44]. Therefore, SCFAs and ketone bodies can reciprocally modulate sympathetic nerve activity via opposing effects on FFAR3 (GPR41), which is abundant in sympathetic ganglia. This is probably part of the overall regulation of body energy expenditure in maintaining metabolic homeostasis that propionate and 3-hydroxybutyrate exert.

It is interesting to note that, although FFAR3 is a Gi/o-coupled GPCR and thus would be expected to act as a sympatho-inhibitory receptor, i.e., as the α_2_ARs and the A1 and A3 adenosine receptors which are also Gi/o-coupled [45], FFAR3 actually stimulates neuronal activity and promotes norepinephrine release from sympathetic nerve terminals (Figure 2) [44]. This is achieved thanks to free Gβγ subunit signaling, rather than through activation of the catalytic Gαi/o subunit, which inhibits AC [45]. Gβγ-dependent signaling that opposes Gαi-dependent inhibition appears to be a more common theme for Gi/o-coupled GPCRs than originally expected or appreciated [46]. In addition, GPCR-kinases (GRKs) and β-arrestins, two major protein families regulating (terminating) GPCR signaling, were not found to be directly involved in FFAR3-induced sympathetic nerve stimulation [44]. Nevertheless, at least their indirect involvement cannot be excluded, given that GRK2 binds and sequesters free Gβγ subunits of activated G proteins at the cell membrane [47] and β-arrestins are known to activate ERK signaling in a G protein-independent manner [48]. Thus, it is quite plausible that GRK2 may oppose PLCβ activation by the free Gβγ subunits of active G proteins and β-arrestins may promote MAPK activation, both of which lead to sympathetic activation and catecholamine release by various receptors [49,50]. These possibilities warrant investigation in future studies.

In conclusion, the ketone body 3-hydroxybutyrate suppresses sympathetic activity and decreases norepinephrine release from sympathetic nerve terminals via FFAR3 (GPR41) blockade (Figure 2). As SGLT2 inhibitors, such as empagliflozin and dapagliflozin, are known to increase production of this ketone body in humans [51], this effect may represent a crucial mechanism via which SGLT2 inhibitors can exert sympatholysis, thereby protecting the failing heart from the deleterious actions of norepinephrine released in excess from cardiac sympathetic nerve terminals during HF.

## 4. Research Hypothesis: Dapagliflozin Attenuates Adrenal Catecholamine Production

At any given time, the human myocardium receives catecholaminergic input, not only from sympathetic nerve terminals releasing norepinephrine directly into the neuro-myocardial junction (Figure 2), but also from the adrenal medulla in the form of (mainly) epinephrine (but also norepinephrine), reaching the heart from the blood after having been released from adrenal chromaffin cells [45]. Both norepinephrine release from sympathetic nerve terminals and catecholamine secretion from the adrenal glands are elevated during the maladaptive sympathetic nervous system hyperactivation that accompanies and aggravates human chronic HF [52,53]. Therefore, both processes need to be suppressed to confer effective sympatholysis during HF treatment. As described in the preceding section, SGLT2 inhibitors may suppress the former process (norepinephrine release from nerve terminals) by upregulating 3-hydroxybutyrate production, which blocks FFAR3-dependent sympathetic nerve activity [44]. This begs the question: can a SGLT2 inhibitor (e.g., dapagliflozin) also affect the latter process (i.e., adrenal catecholamine production)?

To answer this question, we have come up with a novel research hypothesis that centers around the sympatho-inhibitory adrenal α2ARs and their regulation by GRK2.

A plethora of various receptors present in the chromaffin cells of the adrenal medulla promote catecholamine secretion, e.g., neuropeptide Y receptors, muscarinic cholinergic receptors, pituitary adenylate cyclase-activating polypeptide (PACAP) receptors, etc. [54]. In contrast, α2ARs are among the very few (if not the only GPCR type) receptors identified to date that act as inhibitory autoreceptors in adrenal chromaffin cells, inhibiting catecholamine release (Figure 2) [54]. Over the past 15 years, our group and others have established the importance of GRK2 upregulation in the adrenal medulla during HF, which critically regulates catecholamine secretion [52,53,55,56] (Figure 2). Out of a total of seven mammalian GRK isoforms, GRK2 is the most abundant one in the adrenal gland (as it is also in the heart) and is responsible for GPCR phosphorylation and subsequent desensitization, i.e., G protein decoupling [57]. This effectively terminates G protein signaling by the receptor, allowing for subsequent β-arrestin-mediated or other forms of G protein-independent intracellular signaling by the GPCR [48]. Similar to the failing myocardium per se, GRK2 is also overexpressed in the adrenal gland during chronic HF, as a result of the elevated sympathetic nervous system outflow [53,54]. This leads to severe adrenal α2AR dysfunction, as these receptors become significantly desensitized and can no longer suppress CA secretion effectively [52]. Thus, adrenal catecholamine secretion is chronically elevated, contributing to the sympathetic hyperactivity of chronic HF [54,58]. In addition, GRK2 mediates the transcriptional upregulation of tyrosine hydroxylase (TH) in adrenal chromaffin cells (Figure 2) [53]. TH catalyzes the rate-limiting step in catecholamine biosynthesis, i.e., the conversion of tyrosine to L-DOPA (L-3′,4′-dihydroxyphenyalalanine) [54]. The mechanism probably involves activation of the transcription factor CREB (cAMP response element-binding protein) [59,60]. GRK2-mediated TH upregulation results in increased catecholamine (norepinephrine and epinephrine) biosynthesis in the adrenal medulla [53]. Therefore, adrenal GRK2 upregulation induced by the enhanced sympathetic outflow during chronic human HF results in both chronically elevated secretion (via α2AR dysfunction/augmented desensitization) and synthesis of catecholamines. This makes adrenal GRK2 pharmacological targeting an attractive sympatholytic strategy for HF treatment, especially in conjunction with known sympatholytics like the β-blockers [58,61].

Dapagliflozin was recently shown to exert sympatholytic activity in a genetic murine model of hypertension [62]. Among its effects in these animals were suppression of TH expression and of norepinephrine levels in the renal tissue [62]. These findings strongly suggest that dapagliflozin can downregulate TH, thereby inhibiting catecholamine biosynthesis, in tissues [62]. On the other hand, both SGLT1 and -2 have been shown to be present in adrenal gland tissue and, specifically, in adrenal chromaffin cells [63,64]. Taken together, all these findings combined promoted us to hypothesize that dapagliflozin (and potentially other SGLT2 inhibitors, as well) downregulates adrenal GRK2 expression and activity, thereby reducing both synthesis and secretion of catecholamines from the adrenal medulla [65]. Indeed, preliminary data from our laboratory indicate that 24-h-long dapagliflozin treatment leads to significantly reduced GRK2 and TH mRNA and protein levels in the rat chromaffin cell line PC12 (Figure 2) [65]. This, in turn, enhances the inhibitory function of α_2_ARs in chromaffin cells, i.e., effectively decreases epinephrine and norepinephrine secretion tonically stimulated by cholinergic (nicotinic) receptors. At the same time, catecholamine synthesis is also reduced due to the TH downregulation (Figure 2). The potential molecular mechanisms underlying these effects of dapagliflozin are still under investigation, but the pathways at play most likely involve the transcription factors CREB and FOXO1 (forkhead box protein O1). Dapagliflozin has been reported to inhibit the activities of both CREB [66] and FOXO1 [67,68] in vitro and in vivo, although there are a couple of studies arguing for dapagliflozin activating (rather than blocking) FOXO1 [69,70]. These discrepancies in the reported literature are probably the result of the well-known tissue/cell type specific SGLT2 inhibitor’s actions, i.e., as gliflozins can have different effects on the same molecules depending on the cell/tissue type studied. Nevertheless, given that TH is downstream of CREB and GRK2 has been reported to be upregulated by FOXO1, at least in cardiac myocytes [71], it is quite plausible that, in adrenal chromaffin cells, dapagliflozin downregulates both TH and GRK2 via CREB and FOXO1 inhibition, respectively. Of course, as already mentioned, the precise signaling mechanisms involved await delineation, and our efforts to fully elucidate them are under way. Regardless of the underlying pathways however, it is quite likely that dapagliflozin can downregulate TH and GRK2 in the adrenal medulla, leading to a profound sympatholytic effect on this important component of the sympathetic nervous system during chronic human HF.

## 5. Conclusions

Among the beneficial effects of SGLT2 inhibitors in chronic HF is the reduction in sympathetic nervous system activity, irrespective of the concomitant presence of diabetes mellitus. Large clinical trials have already revealed and continue to provide a trove of useful information on the cardiovascular effects of these agents that extend beyond their blood glucose effects. The precise sympatholytic effects of these drugs and their underlying mechanisms are under intense investigation and future research will provide significant additional insights. Nevertheless, the well-established effect of SGLT2 inhibitors at upregulating ketone bodies (mainly β-hydroxybutyrate), which can block the FFAR3-dependent sympathetic neuronal activation, is very likely to prove one of the main mechanisms for the decreased cardiac norepinephrine release/levels this drug class appears to induce in vivo. The present editorial/minireview proposes another such mechanism: SGLT2 inhibitor-induced suppression of adrenal GRK2, which a) restores/enhances sympatho-inhibitory α_2_AR function to decrease adrenal catecholamine secretion, and b) downregulates TH to reduce adrenal catecholamine biosynthesis. If both mechanisms prove to hold true in future studies, then SGLT2 inhibitors will be established as powerful and safe sympatholytic agents in both cardiac noradrenergic nerve terminals and adrenal glands. This is bound to pivot this drug class further ahead in the therapeutic armamentarium for human HF, alongside the current standard, cornerstone HF pharmacotherapies such as β-blockers and renin-angiotensin inhibitors. In any event, the prospects of these agents in the HF pharmacotherapy of the future look much brighter than those of any other drug class currently considered or being developed for the treatment of this devastating disease.

## Figures and Tables

**Figure 1 ijms-22-07684-f001:**
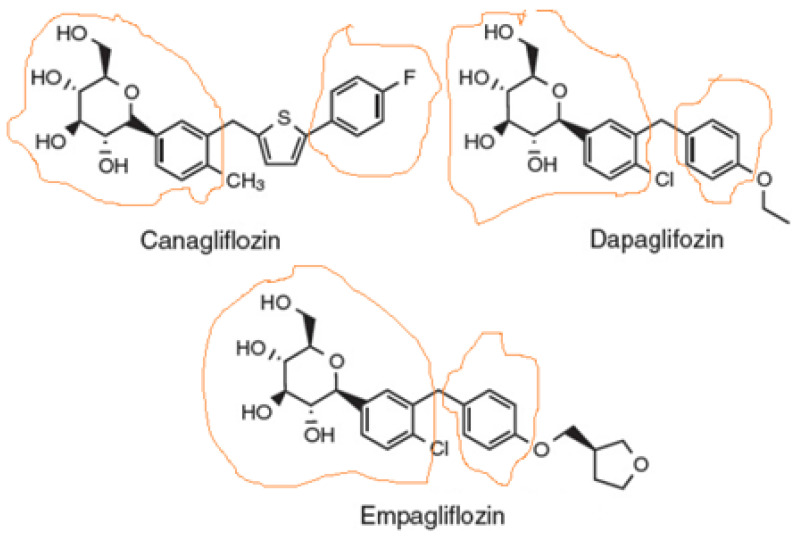
Structures of the three SGLT2 inhibitors currently carrying FDA-approved cardiovascular indications. The parts of the molecule that constitute the pharmacophore are designated as circled by orange lines.

**Figure 2 ijms-22-07684-f002:**
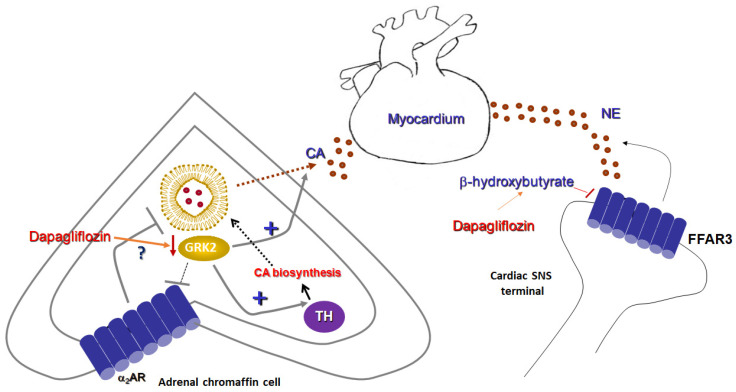
Sympatholytic effects of a SGLT2 inhibitor (dapagliflozin). (**Left**): Proposed sympatholytic mechanism in the chromaffin cell of the adrenal medulla. (**Right**): Potential sympatholytic mechanism in cardiac sympathetic nerve terminals. α_2_AR: Alpha2-adrenergic receptor; CA: Catecholamine; FFAR3: Free fatty acid receptor-3 (GPR41); GRK2: G protein-coupled receptor kinase-2; NE: Norepinephrine; SNS: Sympathetic nervous system; and TH: Tyrosine hydroxylase.

## Data Availability

Not applicable.

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
