# Peer review of "Sympatholytic Mechanisms for the Beneficial Cardiovascular Effects of SGLT2 Inhibitors: A Research Hypothesis for Dapagliflozin’s Effects in the Adrenal Gland"

_ijms, 2021, doi:10.3390/ijms22147684_

Round 1
Reviewer 1 Report
The review manuscript describes about the novel effects of SGLT2 inhibitors on sympatholytic effects in the adrenal. The followings are needed to correct.
1) In lines 51, 52, 53, and 55, the authors describe about the trade name of each SGLT2 inhibitor. Since trade names are different among countries, I do not think it necessary to describe them.
2) In lines 73, 120, 126, 157, 158, 161, 204, 208, 209, 255, 272, 314, and 325, symbol letters are missing. The authors are needed to correct them.
Author Response
We thank this reviewer for the positive comments on the quality of our work.
1) We agree with the reviewer that trade names vary among different countries, so we have removed them from the revised text accordingly.
2) Done. Apologies for this but it happened during the automatic conversion of the text to the journal`s required font style (Palatino). We have corrected all symbols now.
Reviewer 2 Report
Overall, this comprehensive review article is well writing and in a novel concept to review the notion that SGLT2 inhibitors, like dapagliflozin and empagliflozin, exert sympatholytic and outlines the main putative underlying mechanisms for the sympatholytic effects of these drugs in humans, shedding new light on the pharmacological effects that mediate the cardiovascular benefits of SGLT2 inhibitor drugs, independently of their blood glucose-lowering effects. The work is a very nice addition to the field.
Several minor concerns are outlined below.
- Please add more wording on subheadings to help the reader easier understand and gain from each paragraph.
- In the paragraph “3. Current evidence for SGLT2 inhibitor-induced sympatholysis”, please outline 2 to 3 subtitles of evidence to help the reader easier understand.
- There several letters in the text that are missing in lines 73, line 120, line 126, line 133, line 150, lines 157,158, lines 161, line 204, line 208, lines 209, lines243, line 255, line 272, line 314
Author Response
We thank this reviewer for the overall positive comments about the quality of our work.
1) This is a very good suggestion, so we have now added sub-headings to better guide the reader throughout our manuscript. We thank the reviewer for this great recommendation.
2) Again, very good suggestion by this reviewer. We have added a couple of sub-headings in Section #3 of our revised manuscript, as well.
3) As per Reviewer #1`s similar comment, we have now corrected in the revised manuscript all the symbols that were inadvertently missing from our original text due to the automatic conversion of the text to the journal`s required font style (Palatino). Our apologies for this inconvenience.